# Modulation of Th1/Th2 Cytokine Balance by Quercetin In Vitro

**DOI:** 10.3390/medicines7080046

**Published:** 2020-07-30

**Authors:** Yoshihito Tanaka, Atsuko Furuta, Kazuhito Asano, Hitome Kobayashi

**Affiliations:** 1Department of Otolaryngology, School of Medicine, Showa University, Tokyo 142-8555, Japan; ysh10tnk@gmail.com (Y.T.); hitomek@med.showa-u.ac.jp (H.K.); 2Department of Medical Education, School of Medicine, Showa University, Tokyo 142-8555, Japan; atsufuruichi2012@gmail.com; 3School of Health Sciences, University of Human Arts and Sciences, Saitama 339-8555, Japan

**Keywords:** allergic rhinitis, quercetin, human CD4^+^ T cells, Th1/Th2 cytokine balance, modulation, in vitro

## Abstract

**Background:** Allergic rhinitis (AR) is well known to be an IgE-mediated chronic inflammatory disease in the nasal wall, which is primarily mediated by Th2-type cytokines such as IL-4, IL-5, and IL-13. Although quercetin is also accepted to attenuate the development of allergic diseases such as AR, the influence of quercetin on Th2-type cytokine production is not well understood. The present study was designed to examine whether quercetin could attenuate the development of AR via the modulation of Th2-type cytokine production using an in vitro cell culture technique. **Methods:** Human peripheral-blood CD4^+^ T cells (1 × 10^6^ cells/mL) were cultured with 10.0 ng/mL IL-4 in the presence or absence of quercetin. The levels of IL-5, IL-13, and INF-γ in 24 h culture supernatants were examined by ELISA. The influence of quercetin on the phosphorylation of transcription factors NF-κB and STAT6, and mRNA expression for cytokines were also examined by ELISA and RT-PCR, respectively. **Results:** Treatment of cells with quercetin at more than 5.0 μM inhibited the production of IL-5 and IL-13 from CD4^+^ T cells induced by IL-4 stimulation through the suppression of transcription factor activation and cytokine mRNA expression. On the other hand, quercetin at more than 5.0 μM abrogated the inhibitory action of IL-4 on INF-γ production from CD4^+^ T cells in vitro. **Conclusions:** The immunomodulatory effects of quercetin, especially on cytokine production, may be responsible, in part, for the mode of therapeutic action of quercetin on allergic diseases, including AR.

## 1. Introduction

Allergic rhinitis (AR) is well accepted to be a chronic inflammatory IgE-mediated disorder of the nasal wall and is characterized by multiple symptoms such as sneezing, itching, and nasal congestion, among others [1,2]. AR is also accepted to be divided into two different phases of allergic reaction: an initial sensitization phase in which allergen exposure results in IgE formation, and subsequent clinical disease after repeated antigen exposure [3]. The clinical reaction is further subdivided into early- and late-phase responses [1,2]. The development of these responses is orchestrated by Th2-type helper T cells via the production of several types of cytokines and chemokines, which are responsible for the migration and activation of inflammatory cells [1,2].

Current therapeutic agents against AR are limited to antihistamines, antileukotriene, and nasal glucocorticoids that can mitigate allergic symptoms but fail to modulate the allergic reactions and bring adverse side effects such as throat irritation and dry mouth [2,4,5]. Consequently, it is desirable to develop safe and effective therapeutic agents for AR. Quercetin is well known to be one of the most abundant dietary flavonoids, found in various vegetables such as onions, broccoli, tomatoes, etc. [6]. For many years, quercetin has been studied for its possible health benefits, and it has been revealed that quercetin attenuates oxidative stress responses through the suppression of free-radical generation [6,7]; increases in the production of thioredoxin [8] and glutathione [9,10]; quercetin–glutathione conjugate formation [11]; and upregulation of glutamate–cysteine ligases [11], which are important endogenous antioxidants [8,9,10,11]. In regard to allergic immune responses, quercetin has been reported to inhibit the production of both inflammatory cytokines and chemokines such as IL-5, eotaxin, and RANTES (regulated on activation normal T cell expressed and secreted) from eosinophils and mast cells after immunological stimulation in vitro and in vivo [12,13,14,15]. It has also been reported that quercetin inhibits the secretion of harmful chemical mediators, including histamine, leukotrienes, major basic protein, and eosinophil cationic protein from mast cells and eosinophils in vitro and in vivo [14,15]. Furthermore, the influence of quercetin on the production of T-cell cytokines was investigated using an asthmatic mouse model, and it was reported that quercetin could reduce the increased levels of IL-4, and increased IFN-γ levels in bronchoalveolar lavage fluid after antigenic challenge via the modulation of T-box protein expressed in T cells *(T-bet)* and *GATA-3* gene expression, resulting in significant attenuation of all asthmatic reactions [16,17]. Although these reports strongly suggest that quercetin is a good candidate as a supplement for modulation of allergic diseases, including AR, the mechanisms of the therapeutic action of quercetin on allergic responses is not fully understood. There is much evidence that IL-4, one of the Th2-type T-cell cytokines, is a key player in immune modulation of allergic responses and plays essential roles in the development of pathological changes in allergic diseases [2,18]. Although it has been reported that quercetin can inhibit the ability of human peripheral-blood mononuclear cells to spontaneously produce IL-4, but not IFN-γ in vitro via inhibition of cytokine mRNA expression [19], the precise mechanisms of quercetin’ on cytokine production are not well understood. In the present study, therefore, we examined the influence of quercetin on IL-4-mediated immune responses by examining the secretion of cytokines from CD4^+^ T cells in vitro.

## 2. Materials and Methods

### 2.1. Reagents

Quercetin was obtained from Sigma-Aldrich Co., Ltd. (St. Louis, MO, USA) as a preservative-free pure powder. It was dissolved in dimethyl sulfoxide at a concentration of 10.0 mM and was then diluted with RPMI-1640 medium (Sigma-Aldrich Co., Ltd.) supplemented with 10% heat-inactivated bovine serum (RPMI-FBS; Sigma-Aldrich Co., Ltd.) at appropriate concentrations for experiments. It was then sterilized by passing it through 0.2 μm filters, and stored at 4 °C until use. Recombinant human IL-4 was purchased from R & D Systems, Inc. (Minneapolis, MN, USA) as a preservative-free pure powder. IL-4 was also dissolved in RPMI-FBS, sterilized with 0.2 μm filters and stored at 4 °C until use. mRNA isolation kits were purchased from Milteny Biotec (Bergisch Gladbach, Germany). The reagents used for cDNA synthesis and the real-time reverse-transcription polymerase chain reaction (RT-PCR) kit were obtained from Invitrogen Corp. (Carlsbad, CA, USA) and Applied Biosystems (Foster City, CA, USA), respectively.

### 2.2. Preparation of CD4^+^ T Cells

Heparinized human venous blood was obtained from five healthy subjects (all male, 41.0 ± 10.1 years) after obtaining their written informed consent, which was approved by the Ethics Committee of Showa University (Approved No. 190613; Date of approval: 1 June 2019). Peripheral-blood mononuclear cells (PBMCs) were then obtained after centrifugation (1000× *g* for 30 min) of blood with lymphocyte separation medium (Organon Technica, Durham, NJ, USA). CD4^+^ T cells were purified from PBMCs using a magnetic cell separator (Milteny Biotec, Bergisch Gladbach, Germany) as described previously [20]. The cells were suspended in RPMI-FBS at a concentration of 1 × 10^6^ cells/mL. The cell purity was more than 95%, as judged using a flow cytometer (FACScan; Becton Dickinson, San Jose, CA, USA).

### 2.3. Cell Culture

CD4^+^ T cells (1 × 10^6^ cells/mL) were introduced into each well of 24 well culture plates in triplicate, where each well contained 10.0 ng/mL of IL-4 and various concentrations of quercetin in a final volume of 2.0 mL [20]. The supernatants were collected 24 h later and stored at −40 °C until needed for assays for the levels of cytokines. To prepare cells for examining transcription factor activation and mRNA expression, CD4^+^ T cells were cultured in a similar manner for 1 and 4 h, respectively [20]. In all experiments, quercetin treatment was started 1 h before IL-4 stimulation.

### 2.4. Assay for Cytokines

The levels of IL-5, IL-13, and IFN-γ in culture supernatants were measured in duplicate with human cytokine ELISA kits (R & D) according to the manufacturer’s instructions. The sensitivity of the ELISA kits for IL-5, IL-13, and IFN-γ was 3.0 pg/mL, 32.0 pg/mL, and 8.0 pg/mL, respectively.

### 2.5. Assay for Transcription Factor Activities

NF-κB and STAT6 activity in cultured cells were examined using ELISA test kits (Active Mortif Co., Ltd., Carlsbad, Calif, USA) following the manufacturer’s recommended procedures.

### 2.6. Assay for mRNA Expression

Poly A^+^ mRNA was extracted from cells with oligo(dT)-coated magnetic micro beads (Milteny Biotec, Bergisch Gladbach, Germany). mRNA samples (1.0 μg) were reverse-transcribed to cDNA using a Superscript cDNA synthesis kit (Invitrogen Corp., Carlsbad, CA, USA). Polymerase chain reaction (PCR) was then conducted using a GeneAmp 5700 Sequence Detection System (Applied Biosystems, Forster City, CA, USA). The PCR mixture consisted of 2.0 µL of sample cDNA solution (100 ng/µL), 25.0 µL of SYBR-Green Mastermix (Applied Biosystems), 0.3 µL of both sense and antisense primers, and distilled water to give a final volume of 50.0 µL. The reaction was conducted as follows: 4 min at 94 °C, followed by 40 cycles of 4 min at 95 °C, 1 min at 60 °C, and 1 min at 70 °C [20]. GAPDH was amplified as an internal control. mRNA levels for IL-5 and IL-13 were calculated by using the comparative parameter threshold cycle and normalized to GAPDH. The nucleotide sequences of the primers were as follows: for IL-5, 5′-GCTTCTGCATTTGAGTTTGCTAGCT-3′ (sense) and 5′-TGGCCGTCAATGTATTTCTTTATTAAG-3′ (antisense); for IL-13, 5′-CCACGGTCATTGCTCTCAGGCTGGACTG-3′ (sense) and 5′-CCTTGTGCGGGCAGAATCCGCTCA-3′ (antisense) [20]; and for GAPDH, 5′-TGCACCACCAACTGCTTAGC-3′ (sense) and 5′-GGCATGGACTGTGGTCATGAG-3′ (antisense) [7].

### 2.7. Statistical Analysis

Statistical analyses were performed with ANOVA followed by Dunnett’s multiple-comparison test. Values of *p* < 0.05 were considered statistically significant.

## 3. Results

### 3.1. Influence of Quercetin on the Production of T-Cell Cytokines

The first set of experiments was undertaken to examine whether quercetin could suppress the production of Th-2-type cytokines IL-5 and IL-13, by CD4^+^ T cells after IL-4 stimulation. CD4^+^ T cells (1 × 10^6^ cells/mL) were cultured with 10.0 ng/mL IL-4 in the presence of 1.0 to 10.0 μM quercetin for 24 h. The levels of IL-5 and IL-13 in culture supernatants were measured by ELISA. Treatment of cells with quercetin at lower than 2.5 μM did not inhibit IL-5 production: IL-5 levels in experimental culture supernatants were similar (not significant) to those that received IL-4 stimulation alone (Figure 1a). On the other hand, higher concentrations of quercetin (more than 5.0 μM) caused significant suppression of IL-5 production, which was increased by IL-4 stimulation (Figure 1a). We then examined the influence of quercetin on IL-13 production by CD4^+^ T cells after IL-4 stimulation. Quercetin suppressed IL-13 production as it did IL-5 production (Figure 1b). The minimum concentration of quercetin that caused significant suppression of IL-13 production was 5.0 μM (Figure 1b). We finally examined the influence of quercetin on Th-1-type cytokine production using IFN-γ. Stimulation of cells with IL-4 significantly decreased IFN-γ levels in culture supernatants (Figure 2). Although addition of quercetin at less than 2.5 μM did not inhibit the suppressive activity of IL-4 on IFN-γ production, quercetin at more than 5.0 μM suppressed the downregulation of IFN-γ production induced by IL-4 stimulation (Figure 2).

### 3.2. Influence of Quercetin on Transcription Factor Activation and Cytokine mRNA Expression

The final set of experiments was carried out to examine the possible mechanisms by which quercetin could inhibit Th2-type cytokine production from CD4^+^ T cells after IL-4 stimulation. CD4^+^ T cells were stimulated with IL-4 in the presence of 1.0 to 10.0 μM quercetin. Activation of transcription factors NF-κB and STAT6 in 1 h cultured cells was examined by ELISA. As shown in Figure 3a, lower concentrations (1.0 and 2.5 μM) of quercetin did not affect NF-κB activation, which was increased by IL-4 stimulation. However, treatment of cells with higher concentrations (5.0 to 10.0 μM) of quercetin significantly inhibited IL-4–induced NF-κB activation. We then examined the influence of quercetin on STAT6 activation after IL-4 stimulation. The data presented in Figure 3b clearly showed that quercetin inhibited STAT6 activation, as was the case for NF-κB. The minimum concentration of quercetin that caused significant suppression was 5.0 μM. The final experiments in this section were performed to examine the influence of quercetin on Th2-type cytokine mRNA expression in 4 h cultured cells by real-time RT-PCR (Figure 4). Addition of quercetin at 2.5 μM did not suppress mRNA expression for either IL-5 or IL-13, but mRNA expression, which was increased by IL-4 stimulation, was significantly inhibited when cells were treated with quercetin at more than 5.0 μM.

## 4. Discussion

Quercetin, a natural compound belonging to the flavonol subgroup, has been shown to favorably modify the clinical conditions of allergic diseases, including AR, through the inhibition of inflammatory cell (e.g., mast cells and eosinophils) activation [12,13,14,15]. It has also been reported that quercetin exerts suppressive effects on the production of neuropeptides, which are responsible for the development of AR symptoms [21]. Although it is established that Th2-type T cells play a key role in triggering the allergic inflammatory responses in AR [2,18], the influence of quercetin on Th2-type T-cell functions is not clearly defined. The present study, therefore, was undertaken to examine the influence of quercetin on Th2-type T-cell functions by examining Th2-type cytokine production.

The present results clearly showed that quercetin inhibited the ability of CD4^+^ T cells to produce IL-5 and IL-13 after IL-4 stimulation through inhibition of the activation of transcription factors NF-κB and STAT6, and inhibition of cytokine mRNA expression. It is also showed that quercetin abrogated the suppressive activity of IL-4 on INF-γ production by CD4^+^ T cells. The minimum concentration of quercetin that caused significant modulation of cytokine production was 5.0 μM. After oral administration of quercetin at 1200 mg, which is a standard recommended dosage, plasma levels of quercetin gradually increase and peak at 12 μM [22,23], which is a much higher level than that which caused modulation of cytokine production by CD4^+^ T cells after IL-4 stimulation in vitro in this study. From these reports, the findings of the present in vitro study may reflect the biological function of quercetin in vivo.

AR is well known to consist of type I hypersensitivity allergic responses in nasal membranes against several types of aeroallergens [1,2]. It is also accepted that type I allergic responses consist of two different phases [2,3]. The sensitization phase comprises IgE formation against specific allergens based on the Th2-type immune system. In the triggering phase, allergic symptoms are triggered by to secretion of several kinds of chemical mediators from mast cells and eosinophils after re-exposure to the same allergen [2,3] These two phases are orchestrated by T cells, especially Th2-type helper T cells, through the secretion of several cytokines [1,2]. Among Th2-type cytokines, the first important cytokine is IL-4, which promotes the special production of IgE from resting B cells [2,18]. IL-3 and IL-5 are other important Th2-type cytokines, and have been shown to enhance the proliferation and differentiation of mast cells and eosinophils from their precursors [24]. IL-13 is a pleiotropic cytokine produced by activated Th2-type T cells [25]. It has a wide variety of effects on Th2-dominated inflammatory disorders, such as enhancement of IgE production and vascular cell adhesion molecule 1 expression, which increases the migration of inflammatory cells into the site of inflammation [26]. It has also been reported that IL-13 as well as IL-5 can activate and inhibit the apoptosis of eosinophils [25]. On the other hand, IFN-γ, the principal Th1-type effector cytokine, initiates and maintains Th1-type immune responses, which dampen diseases promoted by Th2-type immune responses through the inhibition of Th2-tpe T-cell recruitment/differentiation, induction of apoptosis in eosinophils, and blockage of IgE isotype switch in B cells, among other actions [27]. From these reports, the present results strongly suggest that the beneficial immunomodulatory effects of quercetin may comprise, in part, the therapeutic mode of action of quercetin on allergic diseases, including AR.

Although the present results clearly showed a favorable modification of quercetin on IL-4-mediated Th1/Th2 cytokine balance, the precise mechanism(s) by which quercetin modulates cytokine balance after IL-4 stimulation is not fully understood. IL-4 exerts its biological functions by binding to a high-affinity receptor, IL-4 receptor α chain (IL-4Rα), on the cell surface [28,29], and this complex then induces the activation of the tyrosine kinases, Janus kinase 1 and 3, which cause the phosphorylation of the transcription factor STAT6, which is essential for cytokine production from Th2-type T cells [28,29]. These reports may suggest that the immunomodulatory effect of quercetin on cytokine production is partially dependent on its suppressive activity on the STAT6 signal pathway. This speculation may be supported by the observation that treatment of CD4^+^ cells with quercetin at more than 5.0 μM inhibited STAT6 phosphorylation after IL-4 stimulation. In addition to IL-4Rα, IL-4 binds with the common γ chain and induces phosphorylation of NF-κB, which is responsible for cytokine mRNA expression [30,31]. From these reports, there is another possibility that quercetin inhibits the NF-κB signal pathway and results in suppression of Th2-type cytokine production from CD4^+^ T cells after IL-4 stimulation. This speculation may be supported by the present observation showing the suppressive activity of quercetin at more than 5.0 μM on NF-κB activation induced by IL-4 stimulation.

Activation of Janus kinase 1 and 3 and STAT6 phosphorylation require an increase in intracellular Ca^2+^ levels [32]. Quercetin has been reported to be able to inhibit an increase in intracellular free Ca^2+^ levels in human mast cells after inflammatory stimulation in vitro [33]. Quercetin has also been reported to inhibit the phosphorylation of several types of tyrosine kinases, which are responsible for transcription factor activation [34,35]. On the basis of these reports, quercetin might inhibit the phosphorylation of tyrosine kinases through the inhibition of an increase in Ca^2+^ levels in CD4^+^ cells after IL-4 stimulation, resulting in suppression of Th2-type cytokine production. Further experiments are required to clarify this point.

## 5. Conclusions

The present results strongly suggest that quercetin modulates IL-4-mediated immune responses, especially Th1/Th2 cytokine balance, and results in attenuation of the development of allergic immune responses.

## Figures and Tables

**Figure 1 medicines-07-00046-f001:**
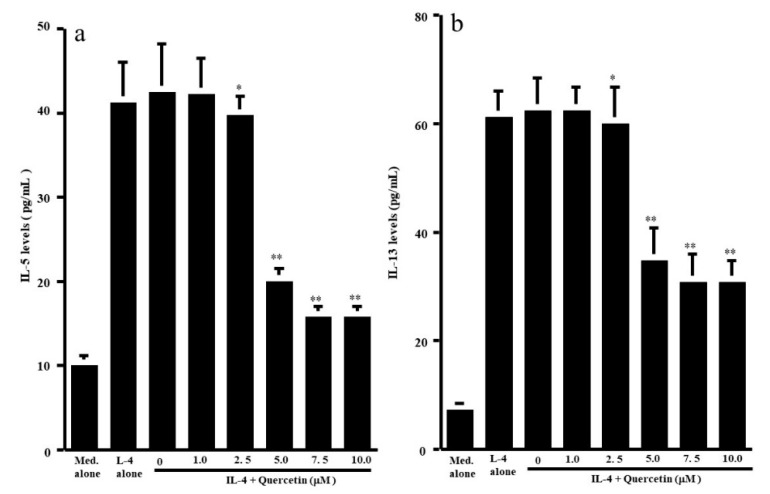
Influence of quercetin on Th2-type cytokine production from human peripheral-blood CD4^+^ T cells in vitro. CD4^+^ T cells (1 × 10^6^ cells/mL) were stimulated with 10.0 ng/mL IL-4 in the presence of various concentrations of quercetin for 24 h. Cytokine levels in culture supernatants were examined by ELISA. The results were expressed as the mean pg/mL ± SE of five subjects. (**a**): IL-5; (**b**): IL-13; * *p* > 0.05 versus IL-4 alone; ** *p* < 0.05 versus IL-4 alone.

**Figure 2 medicines-07-00046-f002:**
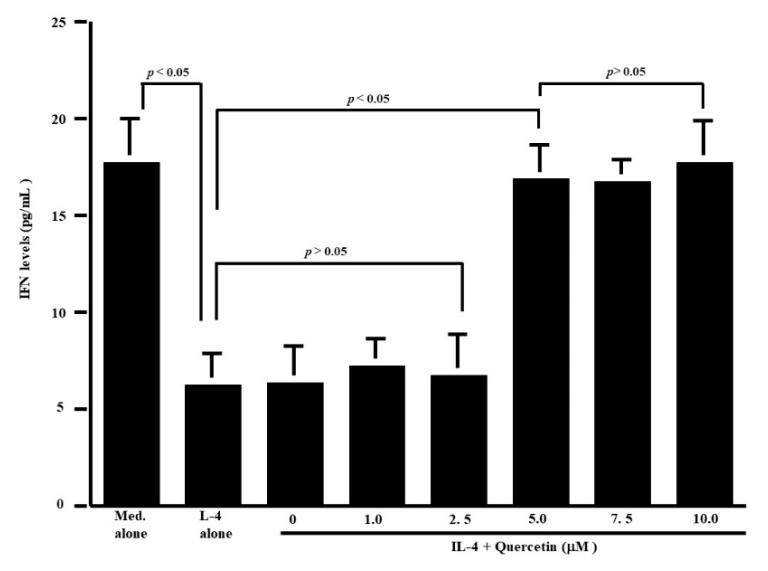
Influence of quercetin on interferon (IFN)-γ production from human peripheral-blood CD4^+^ T cells in vitro. CD4^+^ T cells (1 × 10^6^ cells/mL) were stimulated with 10.0 ng/mL IL-4 in the presence of various concentrations of quercetin for 24 h. IFN-γ levels in culture supernatants were examined by ELISA. The results were expressed as the mean pg/mL ± SE of five subjects.

**Figure 3 medicines-07-00046-f003:**
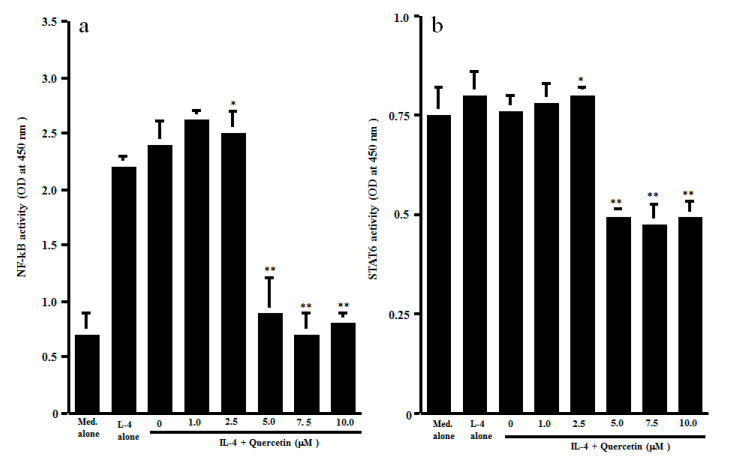
Influence of quercetin on transcription factor activation in CD4^+^ T cells in vitro. CD4^+^ T cells (1 × 10^6^ cells/mL) were stimulated with 10.0 ng/mL IL-4 in the presence of various concentrations of quercetin for 1 h. Activation of transcription factors NF-κB (**a**) and STAT6 (**b**) was assessed by ELISA. The results were expressed as the mean OD at 450 nm ± SE of five subjects. * *p* > 0.05 versus IL-4 alone; ** *p* < 0.05 versus IL-4 alone.

**Figure 4 medicines-07-00046-f004:**
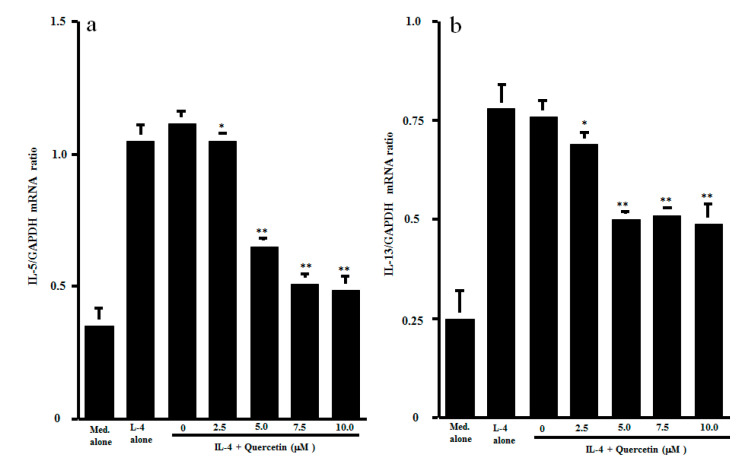
Influence of quercetin on mRNA expression for Th2-type cytokines in vitro. CD4^+^ T cells (1 × 10^6^ cells/mL) were stimulated with 10.0 ng/mL IL-4 in the presence of various concentrations of quercetin for 4 h. mRNA expression for IL-5 (**a**) and IL-13 (**b**) was examined by real-time RT-PCR. The results were expressed as the mean cytokine/GAPDH ± SE of five subjects. * *p* > 0.05 versus IL-4 alone; ** *p* < 0.05 versus IL-4 alone.

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
