# Peer review of "Modulation of Th1/Th2 Cytokine Balance by Quercetin In Vitro"

_medicines, 2020, doi:10.3390/medicines7080046_

Round 1

Reviewer 1 Report

Quercetin has demonstrated significant effects on immune system response and direct inhibition of several processes of inflammation as has been shown by the following articles, among many others:

Huang R-Y et al. Immunosuppressive effect of quercetin on dendritic cell activation and function. J Immunol 2010;184:6815-21.

Sternberg Z et al. Quercetin and interferon-b modulate immune response (s) in peripheral blood mononuclear cells isolated from multiple sclerosis patients. J Neuroimmunol. (2008) 205:142-147.

Hee-ju Park H-J et al. Quercetin Regulates Th1/Th2 balance in a murine model of asthma. Int Immunopharmacol 2009;9(3):261-267.

Rogerio AP et al. Anti-inflammatory effect of quercetin-loaded microemulsion in the airways allergic inflammatory model in mice. Pharmacol Res 2010;61:288-297.

Insani E et al. TH1/TH2 balance in intestinal fluid after oral administration of short-chain fructo-oligosaccharides and quercetin. Proceedings of the Nutrition Society 2011;70(OCE2), E38.

Ravikumar N and Kavitha CN. Immunomodulatory effect of Quercetin on dysregulated Th1/Th2 cytokine balance in mice with both type 1 diabetes and allergic asthma. Journal of Applied Pharmaceutical Science 2020;10(3):80-87.

Hosseinzade A et al. Immunomodulatory Effects of flavonoids: possible induction of T CD4+ regulatory cells through suppression of mTOR pathway signaling activity. Frontiers in Immunology 2019;10:1.

Chirumbolo S. The role of quercetin, flavonols and flavones in modulating inflammatory cell function. Inflamm Allergy Drug Targets 2010;9:263-285.

Park HJ et al. Quercetin regulates Th1/Th2 balance in a murine model of asthma. Int Immunopharmacol 2009;9:261-267.

In the submitted paper, Tanaka et al. examined the effect of quercetin on IL-4-stimulated human CD4+ T cells by measuring IL-5, IL-13 and IFN-γ levels, NF-B and STAT6 activation, and mRNA levels for IL-5 and IL-13 and GAPDH. The paper is based on the fact of current investigations on the development of safe and effective therapeutic agents for inflammatory conditions as well as to the fact that the mechanisms by which quercetin regulates cytokine production are not fully understood.

The results are shown in 4 figures. It was shown in IL-4-exposed cells that quercetin inhibits the IL-5 and IL-13 mRNA expression as well as the production of IL-5 and IL-13 and increases IFN-g production through inhibition of NF-B and STAT6 transcription factors activation.

This is well-designed and performed study. The conclusions are supported by the data presented.

Minor observations

Line 43. and tomatoes, etc [6].

(Please correct to “etc.”

Line 47. chemical mediators

(Please be more specific).

Line 45. secretion of thioredoxin, the most important endogenous antioxidant [8].

(Please consider other important endogenous antioxidants as glutathione, lipoic acid, coenzyme Q, ferritin, bilirubin, the core of antioxidants enzymes, etc.).

Line 155. Transcription factor, NF-155 B and STAT6, activation

(please use the plural for “transcription factor”).

Line 181: through the inhibition of inflammatory cell (e.g. mast cells and eosinophils

(please use the plural for “inflammatory cell”).

Figures:

“Y” axis titles of figures should be indicating the parameters and measurements units only; the mean and dispersion are not necessary in axis text.

Author Response

Referee 1

I greatly appreciate your valuable comments. My replies to your specific comments are as follows. Revised portions are marked with red ink.

Major comment:

Comment 1: Qercetin has demonstrated significant effects on immune system responses and direct inhibition of several processes of inflammation as has been shown by the following articles.

Response: According to this comment, I revised the introduction section (Lines 51 to 60). To do this, 2 new references were added (No. 16 and 17).

Other comments:

Comment 1: Correct miss spelling.

Response: I carefully read the manuscript and corrected typographical errors.

Comment 2: Please consider other important endogenous antioxidant as glutathion----.

Response: According to these comments, I revised the introduction section (Lines 49 to 51).

To do this, 3 new references were added (No. 9, 10, and 11).

Comment 3: Change “Y” axis title of figures.

Response: According to this comment, I revised title of “Y” axis in figures (Figures 1 to 4).

Other changes:

  1. According to the comment raised by Referee 2, I deleted the sentences showing the clinical symptoms of AR.

    2. According to the iThenticate report, I carefully read the entire manuscript           and revised it as much as possible.

Reviewer 2 Report

Nice piece of works in vitro for precise analysis of Quercetin, which have been reported to have an inhibitory effect on allergic inflammation.

Data is OK to be understood and well-written manuscript with in vitro study.

However, authors should employ allergic rhinitis model in vivo and investigate with quercetin effect for attenuating nasal symptoms of AR and analyse cytokine modulation of it. 

Even though they mentioned other report on clinical effect of it, but it does not relate with cytokine modulation. 

Otherwise, authors should not refer nasal symptoms of AR and delete these sentences in this manuscript. 

Author Response

Referee 2

I greatly appreciate your valuable comments. My replies to your specific comments are as follows.

Revised portions are marked with red ink.

Major comments

Comment 1: Authors should employ allergic rhinitis model in vivo and investigate with quercetin effects for

attenuating nasal symptoms on AR and analyse cytokine modulation of it.

Response: Thank you for your excellent advice. We are now planning to examine the in vivo effect of

quercetin on cytokine balance by using experimental mouse model of AR.

Comment 2: Authors should not refer nasal symptoms of AR and delete these sentences in the manuscript.

Response: According to this comment, I revised the manuscript (Introduction section and Conclusion).

Other changes:

1) According to the comments raised by Reviewer 1, I revised the introduction section (Lines 49 to 60). To do this, 5 new references were added (No. 9, 10, 11, 16, and 17).

2) According to the iThenticate report, I carefully read the entire manuscript and revised it as much as possible.

Round 2

Reviewer 2 Report

ok to be accepted for publication.

well revised  to delete overdiscussion.